# Experimental Study on the Dissolution Behavior of Calcium Fluoride

**Suchandra Sar \*** , **Caisa Samuelsson, Fredrik Engström and Lena Sundqvist Ökvist**

Division Minerals and Metallurgical Engineering, Department of Civil, Environment and Natural Resources Engineering, Luleå University of Technology, 971 87 Luleå, Sweden; caisa.samuelsson@ltu.se (C.S.); fredrik.i.engstrom@ltu.se (F.E.); lena.sundqvist-oqvist@ltu.se (L.S.Ö.)

\* Correspondence: suchandra.sar@ltu.se; Tel.: +46-72-708 4245

**Abstract:** The presence of halogens has an adverse effect on the zinc extraction process through electrowinning, the last phase of the RLE (Roasting, Leaching and Electrowinning) zinc extraction route. Fluoride ($F^-$) may be present as calcium fluoride ($CaF_2$) and this is, for example, the case in double leached Waelz oxide (DLWO). Efficient removal of $F^-$ from primary and secondary raw materials for zinc extraction results in a simplified process and increases flexibility in the selection of raw materials. Understanding of the solubility behavior of pure $CaF_2$ can give valuable information on treatment for maximized halogen removal. Dissolution of $CaF_2$ was studied with the addition of sodium carbonate ($Na_2CO_3$) and sodium bicarbonate ($NaHCO_3$). Dissolution studies were combined with thermodynamic calculations to understand the solubility behavior of $CaF_2$ under different conditions. Results from the experiments and the thermodynamic calculations show that $Na_2CO_3$ and $NaHCO_3$ have similar behavior if the pH is controlled at the same value. The available carbonate ($CO_3^{2-}$) ion in the system limits the concentration of calcium ($Ca^{2+}$) ion by precipitation of $CaCO_3$, which enhances the dissolution of $CaF_2$. At higher temperatures and pH, calcite, vaterite, and aragonite were formed and co-precipitation of $CaF_2$ along with calcium carbonate ($CaCO_3$) was observed. At lower temperatures and lower pH levels, only calcite and vaterite were formed and a coating by $CaCO_3$ on $CaF_2$ was found to hinder complete dissolution reaction. The results of this study indicate that the temperature along with the reagents used for the dissolution tests have a significant impact on the $CaCO_3$ polymorph mixture (calcite, vaterite and aragonite) formation.

**Keywords:** dissolution of calcium fluoride; calcium carbonate polymorphism; dehalogenation; double leached Waelz oxide; effect of fluoride during electrowinning

## 1. Introduction

Due to increased use of zinc (Zn), the worldwide production and the demand for zinc-bearing raw materials are increasing [1]. Extraction of Zn from secondary resources such as Electric Arc Furnace (EAF) dust reduces the dependency on primary ore and landfilling can be avoided. Globally, 80% of Zn is produced via the roasting, leaching and electrowinning (RLE) route. EAF dust with Zn contents of 4–33% is treated in a Waelz kiln at temperatures up to 1250 °C in CO-$CO_2$ reducing atmosphere to produce Zn enriched dust known as Waelz oxide (WO), containing roughly ~60% of Zn [2]. WO is further treated for dehalogenation [3–5] forming double leached Waelz oxide (DLWO). DLWO contains 61–76% Zn, 1.6–9.0% lead and 1.5–5.5% iron. The contents of chloride ($Cl^-$) and fluoride ($F^-$) are 0.10–0.16% and 0.10–0.15%, respectively [6,7].

During electrowinning, $F^-$ attacks the aluminum (Al) cathode, leading to formation of cavities where Zn deposits. The deposited Zn may form an alloy with Al cathode, which leads to difficulties in stripping the Zn sheet [8–11].

Of the few methods of removal of halogens from WO, the most common is through lixiviation with sodium carbonate ($Na_2CO_3$) [3] at pH 8–10 (Refer Appendix A.1 about self-generated pH of WO slurry) and temperatures of 50–90 °C [3,5]. Autoclave pressure leaching of WO by using $Na_2CO_3$ is also an industrially established process for removal of the halogens [12] WO can, alternatively, be washed in a single stage leaching with 100 g/L $Na_2CO_3$, at 90 °C with pulp density of 30% for 3 h [5]. Characterization studies conducted on DLWO confirmed the presence of calcium fluoride ($CaF_2$) [4,7]. Furthermore, $F^-$ content determined with fluoride ion selective electrode (FISE), $^{19}F$ Liquid State Nuclear Magnetic Resonance (LS NMR) and calcium content determined by inductively coupled plasma-optical emission spectroscopy (ICP-OES) showed that the content of $CaF_2$ was ~0.2% [7] in analyzed samples. Though literature on leaching of WO [3–5] with $Na_2CO_3$ gives a general explanation on removal of halides, detailed knowledge of the dissolution behavior of $CaF_2$ during applied alkaline leaching conditions is not available.

The alkaline leaching [5] can be described by the following reaction:

$$MeF_2 \text{ (s)} + Na_2CO_3 \text{ (aq)} \rightarrow MeCO_3 \text{ (s)} + 2\,NaF \text{ (aq)} \tag{1}$$

where Me = Zn, Pb, Ca.

It is known from studies conducted on natural ground water at alkaline pH of 7.5–8.5 due to naturally occurring sodium bicarbonate, dissociation of natural $CaF_2$ is enhanced, precipitating calcium carbonate ($CaCO_3$) as calcite and increasing $F^-$ concentration in ground water [13–15]. Based on this knowledge the solubility of fluorite was studied by Rao et al. [15] in the laboratory at 28 °C with 1M $Na_2CO_3$ and 2M $NaHCO_3$ as leaching reagent precipitating calcite after 30 days of leaching. The dissociation of $CaF_2$ is expressed as:

$$CaF_2(s) + 2NaHCO_3 \text{ (aq)} \rightarrow CaCO_3 \text{ (s)} + 2NaF + 2H^+ + CO_3^{2-} \tag{2}$$

$CaCO_3$ is known to exhibit polymorphism i.e., the ability of crystalline system to exist in more than one crystal structure. A review article by Meldrum, F.C., & Cölfen, H. on polymorphism of $CaCO_3$ on biological and synthetic system indicates three crystalline polymorphs of anhydrous $CaCO_3$ namely vaterite, aragonite, and calcite, listed here in order of increasing thermodynamic stability [16]. In nature, aragonite and calcite are commonly seen. The least dense amorphous $CaCO_3$ is precipitated first, which rapidly changes to hollow metastable vaterite, which further changes to aragonite or calcite depending on the free energies of activation of nucleation in different environments. Several experimental studies were conducted to understand $CaCO_3$ polymorphism using soluble salts of calcium where the sources of carbonate ions include soluble carbonates and bicarbonates [16,17]. However, literature on polymorphism exhibited by precipitated $CaCO_3$ from sparingly soluble $CaF_2$ is not available. Ichikuni (1979) showed that due to ion exchange reaction, one carbonate ion in aragonite is replaced by two fluoride ions, leading to coprecipitation of fluoride in $CaCl_2$-$Na_2CO_3$-$NaF$ system [18] Y. Kitano and M. Okumura concluded that the coprecipitation of fluoride with aragonite is larger than calcite [19].

Therefore, in order to develop an efficient removal method of $CaF_2$ from secondary zinc-containing dust material such as WO and DLWO, understanding the dissolution chemistry of $CaF_2$ with $Na_2CO_3$/$NaHCO_3$ and the crystallization process of $CaCO_3$ in a more detailed manner is required. For this purpose, a dissolution study of pure $CaF_2$ was conducted under different conditions with either $Na_2CO_3$ or $NaHCO_3$ as reagent. Results from laboratory dissolution tests were evaluated with support from thermodynamic calculation using FactSage 7.2. Design of $CaF_2$ dissolution tests was based on reported research work and patents [3,5,12] and on the fact that the dissolution of zinc oxide is lowest at pH around 9 [4].

## 2. Materials and Methods

### 2.1. Reagents

Chemicals used are calcium fluoride ($CaF_2$ 99.5% AlfaAesar GmbH & Co, Karlsruhe, Germany), anhydrous sodium carbonate ($Na_2CO_3$ 99.5%, AlfaAesar, Thermo Fisher-Kandel GmbH, Germany),

sodium hydrogen carbonate ($NaHCO_3$ 99%, AlfaAesar, Thermo Fisher-Kandel GmbH, Germany), total ionic strength adjustment buffer (TISAB, VWR International bvba, Leuven, Belgium), sodium fluoride (NaF EMSURE, Merck KGaA, Darmstadt, Germany), nitric acid ($HNO_3$ 68%, VWR International S.A.S, Fontenay-sous-Bois, France) and 10 M sodium hydroxide (NaOH) solution prepared by standard method using NaOH pellets (EMPROVE ESSENTIAL, Merck KGaA, Darmstadt, Germany).

## 2.2. Particle Size

Particle size distribution of $CaF_2$ was conducted through laser-diffraction measurements using a CILAS 1064 unit (CILAS, Orleans, France) and Fraunhofer approximation was used for the evaluation of the collected data.

## 2.3. Dissolution Experiments

The dissolution experiments were conducted batch-wise in a four-necked round-bottomed 1-L glass reactor fitted with a dropping funnel, mechanical paddle stirrer, pH electrode, thermometer, and heating mantle. The suspensions were stirred at a speed of 300 rpm/min and the temperature was kept at set value by using a temperature controller. The experimental dissolution conditions were: reagents (150 mM $Na_2CO_3$ and 150 mM $NaHCO_3$); temperatures (70 °C and 95 °C) and pH (10.6 and 9.0). The natural pH of 150 mM (16 g/L) $Na_2CO_3$ is 10.6 and of 150 mM (12.68 g/L) $NaHCO_3$ is 8.7. To meet the need to decrease or increase the pH, concentrated $HNO_3$ or 10 M NaOH, respectively, was added. One liter of water with selected dissolution reagent was taken in a glass reactor, heated to desired temperature, pH was adjusted and solid 300 mg/L $CaF_2$ added. To understand the reaction kinetics, experiments were performed for 480 min and, at selected time intervals, 20 mL samples of the solution were collected for the analysis of dissolved $F^-$ in the leach liquor. To minimize the error due to large sampling volume for each dissolution condition, three identical experimental set ups were assembled. From each experimental set up, only three samples were collected but at different pre-determined time intervals (after 5 min, 15 min, and 30 min from set up one; after 60 min, 120 min, and 180 min from set up two; after 240 min, 360 min, and 480 min from set up three). On completion of the experiment, the solid residue from set three was filtered and dried for mineralogical characterization by XRD-Rietveld refinement and SEM-EDS.

## 2.4. Fluoride Analyses

Fluoride analyses were carried out using Fluoride Ion Selective Electrode (FISE Metrohm 6.0502.150, Herisau, Switzerland, Titrando 888 with Tiamo software) and the standard methods for the examination of water and wastewater were applied [20]. A calibration curve with measured potentials in mV as function of concentration in mg/L was plotted for standards of, 2, 20, and 200 mg/L. The content of $F^-$ was deduced from the calibration curve using the measured potentials in mV for each sample.

## 2.5. Thermodynamic Calculations

The Equilibrium module of FactSage 7.2 (Thermfact/CRCT, Montreal, QC, Canada and GTT-Technologies, Aachen, Germany) [21,22] was used to carry out equilibrium calculations in the pH range of 7 to 11 by changing the concentration of $H^+$ ion and using the database FactPS (Fact Pure substances database 2018). Calculations involving aqueous species such as $Na^+$, $HCO_3^-$, $CO_3^{2-}$, $NaHCO_3$, $F^-$, $Ca^{2+}$ and gaseous species $CO_2$ were conducted for the experimentally used dissolution conditions. This also included the effect from temperature, type of reagent, the added amount of reagent and $CaF_2$.

## 2.6. Inductively Coupled Plasma-Optical Emission Spectroscopy

Inductively Coupled Plasma-Optical Emission Spectroscopy (ICP-OES, Model: Thermo Scientific ICAP 7200 DUO, Thermo Fisher Scientific, Waltham, MA, USA) was used to analyze the calcium and sodium content in the solution.

*2.7. Quantitative X-ray Powder Diffraction with Rietveld Analysis*

To determine and quantify mineralogical components present in the precipitated residue from the dissolution experiments X-ray diffraction (XRD) analyses were performed. A PANalytical Empyrean X-ray diffractometer (Malvern Panalytical, Almelo, The Netherlands), equipped with copper $K_\alpha$ radiation of 45 kV and 40 mA was used for XRD analyses. XRD patterns were recorded in the 2-theta range 10–90° with a step of 0.026°. Evaluation of data was carried out using the software HighScore Plus and the Inorganic Crystal Structure Database (ICSD). For Rietveld refinement, background correction, scale factors, unit cells, preferred orientations and profile variables were included.

*2.8. Scanning Electron Microscopy (SEM) Coupled with Energy Dispersive Spectroscopy (EDS)*

SEM-EDS was conducted on the polished and carbon-coated epoxy sample using Zeiss Gemini Merlin equipped with an energy dispersive spectrometer (EDS-Xmax 80 mm, Zeiss, Oberkochen, Germany). The acceleration voltage was set to 10 kV and the emission current was 250 pA.

## 3. Results and Discussion

*3.1. Particle Size*

The particle size distribution of $CaF_2$ feed showed that the mean diameter was 3.37 μm, 50% particles were <3.00 μm, whereas the majority (90%) of the particles were <5.93 μm. All the particles are less than 13 μm in size.

*3.2. Calcium Fluoride Dissolution and Thermodynamic Calculations*

3.2.1. Dissolution at pH 10.6

Dissolution of $CaF_2$ for 480 min at 70 °C and pH 10.6 with $Na_2CO_3$ showed 99.3% dissolution of fluoride whereas at 95 °C and pH 10.6 83.3% dissolution was achieved (Figure 1). At higher temperature (95 °C) the dissolution reaction is initially rapid and $F^-$ content in the solution reaches maximum after approximately 100 min. The dissolution rate is retarded after 100 min. As can be seen from Figure 1, similar result was observed when $NaHCO_3$ was used and the pH was adjusted to 10.6 using 10 M NaOH. However, an equilibrium calculation based on experimental data indicates that complete dissolution of added amount of $CaF_2$ is theoretically possible under the given experimental condition (Figure 1).

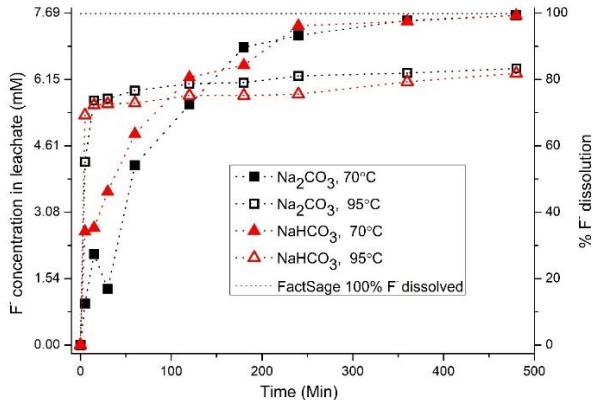

**Figure 1.** Experimental data for dissolution of $CaF_2$, at pH 10.6 as function of time in comparison with thermodynamically calculated concentration based on the experimental conditions.

### 3.2.2. Dissolution at pH 9.0

Figure 2 shows the development of dissolution of $CaF_2$ over time at pH 9.0. With $NaHCO_3$ at 70 °C, the $CaF_2$ dissolution was 88.3% after 480 min whereas at 95 °C it was 96.7%. Similar results were obtained with $Na_2CO_3$ at 95 °C when the pH was adjusted to pH 9.0 using $HNO_3$; at 70 °C the dissolution was slightly lower compared to that of $NaHCO_3$. At low temperature and low alkaline pH the rate of the reaction is slower. At 70 °C, the % dissolution of fluoride increases with increase in time. Compared to at pH 10.6 the increase in initial % dissolution is less at pH 9.0, but the final % dissolution reached at pH 9 is 13% higher at 95 °C. Again, equilibrium calculation based on experimental data indicates that complete dissolution of added amount of $CaF_2$ is theoretically possible under the given experimental conditions (Figure 2).

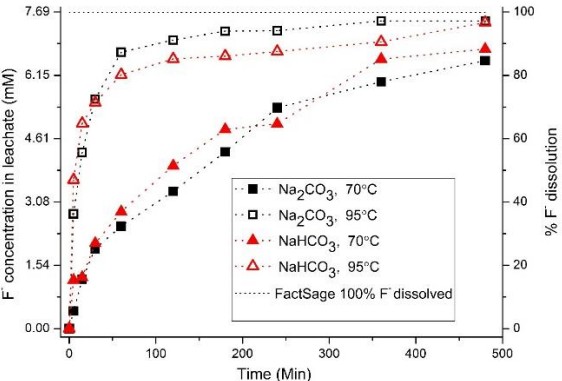

**Figure 2.** Experimental data for dissolution of $CaF_2$ at pH 9.0 as a function of time in comparison with thermodynamically calculated concentration based on the experimental conditions.

### 3.2.3. Thermodynamic Calculations

In Table 1, thermodynamically calculated and experimentally found ionic concentrations (mM) for existing species, at conditions corresponding to the experimental ones, possibly influencing dissolution are compared. The major specie in the carbonate system at pH 9.0 is $HCO_3^-$ and at pH 10.6 it is $CO_3^{2-}$. Thermodynamically calculated (Refer Figures A1 and A2) and experimentally determined concentrations of $Na^+$ and $F^-$ are comparable but $Ca^{2+}$ concentration differs three orders in magnitude (in the case of $NaHCO_3$) and four orders in magnitude (in the case of $Na_2CO_3$). According to the literature [23,24], the solubility product of $CaF_2$ does not differ significantly between 70 and 95 °C. Due to kinetic reasons, complete dissolution as predicted by equilibrium calculation is being hindered.

**Table 1.** Thermodynamically calculated and experimentally determined concentrations (mM) of aqueous species with 300 mg/L $CaF_2$ and 150 mM of dissolution agent.

| Dissolution Condition | Thermodynamically Calculated in mM | | | | | | Experimentally Determined in mM | | |
|---|---|---|---|---|---|---|---|---|---|
| | $CO_3^{2-}$ | $HCO_3^-$ | $NaHCO_3$ | $Na^+$ | $Ca^{2+}$ | $F^-$ | $Na^+$ ICP-OES | $Ca^{2+}$ ICP-OES | $F^-$ FISE |
| $Na_2CO_3$, 70 °C, pH 10.6 | 97.1 | 36.5 | 7.97 | 283 | $1.20 \times 10^{-5}$ | 7.68 | 281 | $5.1 \times 10^{-2}$ | 7.63 |
| $Na_2CO_3$, 95 °C, pH 10.6 | 94.6 | 39.9 | 6.17 | 283 | $0.49 \times 10^{-5}$ | 7.68 | 286 | $3.9 \times 10^{-2}$ | 6.40 |
| $NaHCO_3$, 70 °C, pH 9.0 | 8.39 | 126 | 12.9 | 138 | $14.9 \times 10^{-5}$ | 7.68 | 173 | $3.0 \times 10^{-2}$ | 6.78 |
| $NaHCO_3$, 95 °C, pH 9.0 | 7.12 | 120 | 9.98 | 150 | $6.30 \times 10^{-5}$ | 7.68 | 163 | $2.1 \times 10^{-2}$ | 7.44 |

### 3.3. Mineral Phases in Residue by XRD

The diffractograms of the residues collected after 480 min dissolution are presented in Figure 3. Phases detected were different polymorphs of $CaCO_3$ such as calcite, vaterite and aragonite in different proportions. When dissolution of $CaF_2$ was done with $Na_2CO_3$ at 70 °C, calcite and vaterite were detected in the residue along with 1.5% fluorite ($CaF_2$) (Table 2). When the reaction temperature is higher (95°) 15% aragonite is also present along with 15.2% fluorite in the residue. According to the

literature, aragonite formation is favored when the precipitation reaction is rapid [18] and at higher pH [25]. At a temperature of 95°, the dissolution of $CaF_2$ is initially rapid (Figure 1), indicating that precipitation of $CaCO_3$ may also be rapid. Aragonite formation at 95° leads to more coprecipitation of fluoride compared to calcite [18], inhibiting complete dissolution of fluoride.

In the diffractograms of residues from dissolution with $NaHCO_3$ at pH 9.0 and temperature 70 and 95°, calcite was found to be the dominating phase but aragonite (Table 2) was also observed in the residue from dissolution at 95° (Figure 3). Also, for this sample, the initial rate of the dissolution is rapid (Figure 2) and this leads to rapid precipitation of $CaCO_3$. In the residue obtained from dissolution experiment with $NaHCO_3$ at pH 9.0 and 70 °C, no aragonite was found but 9.1% of $CaF_2$ was found.

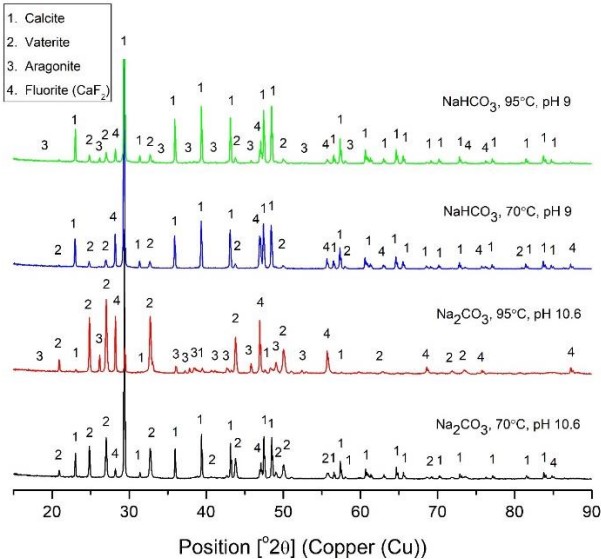

**Figure 3.** XRD diffractogram of residues after dissolution reaction of 300 mg/L $CaF_2$ for 480 min.

As can be seen in Table 2, the highest total amount of $CaCO_3$ is present in the solid residue after dissolution with $Na_2CO_3$ at 70 °C and pH 10.6, $CaCO_3$ is present as calcite and vaterite. Almost as much $CaCO_3$, but mainly calcite, is formed with $NaHCO_3$ dissolution done at 95 °C and pH 9.0.

**Table 2.** Amount of present minerals based on residue analysis by XRD-Rietveld refinement and calculated considering total amount of residue generated.

| Residue (Dissolution Condition) | Calcite | | Vaterite | | Aragonite | | Total CaCO₃ | | Fluorite | |
|---|---|---|---|---|---|---|---|---|---|---|
| | mg | % | mg | % | mg | % | mg | % | Mg | % |
| $Na_2CO_3$, 70 °C, pH 10.6 | 200 | 56.2 | 151 | 42.4 | - | - | 351 | 98.6 | 5.30 | 1.50 |
| $Na_2CO_3$, 95 °C, pH 10.6 | 26.8 | 7.70 | 216 | 62.1 | 52.2 | 15.0 | 295 | 84.8 | 52.9 | 15.2 |
| $NaHCO_3$, 70 °C, pH 9.0 | 254 | 80.4 | 33.2 | 10.5 | - | - | 288 | 90.9 | 28.8 | 9.10 |
| $NaHCO_3$, 95 °C, pH 9.0 | 294 | 82.7 | 39.8 | 11.2 | 11.0 | 3.00 | 345 | 96.9 | 10.7 | 3.10 |

According to Jun Kawano et al. [26], the formation of different polymorphs or mixture of polymorphs is dependent on the precipitation temperature. Researchers A. Sarkar and S. Mahapatra [17] reported transformation of metastable vaterite to thermodynamically more stable aragonite or even more stable calcite depending on the conditions. Thermodynamically, the dissolution of vaterite leading to the formation of aragonite is possible, but this kind of transformation is rarely observed. There is a kinetic barrier for this transformation that may take place at elevated temperature when the activation energy is supplied. The highest amount of vaterite (Table 2) was formed when dissolution was done at pH 10.6 with $Na_2CO_3$ at 95 °C and in the same dissolution experiment the highest amounts of aragonite and fluorite were found. Initial rapid dissolution kinetics leads to rapid precipitation

and, hence, the least thermodynamically stable vaterite was formed. Less thermodynamically stable crystals are formed when crystallization process is kinetically controlled.

Relative to the initial amount of $CaF_2$ added (300 mg/L), %$CaF_2$ dissolved was estimated based on $F^-$ content in the solution and %$CaF_2$ in the residue estimated via the Rietveld method applied on XRD data, both estimations are stated in Table 3. After 8 h of reaction the highest dissolution (Table 3) of $CaF_2$ was found at 70 °C and pH 10.6 with $Na_2CO_3$.

**Table 3.** %$CaF_2$ dissolved based on %F in solution and present in residue, both calculated considering the feed weight (300 mg/L $CaF_2$).

| Dissolution Condition | % $CaF_2$ Dissolved in Solution-FISE | % $CaF_2$ in Residue XRD |
|---|---|---|
| $Na_2CO_3$ at 70 °C, pH 10.6 | 99.3 | 1.8 |
| $Na_2CO_3$ at 95 °C, pH 10.6 | 83.3 | 17.6 |
| $NaHCO_3$ at 70 °C, pH 9.0 | 88.3 | 9.6 |
| $NaHCO_3$ at 95 °C, pH 9.0 | 96.7 | 3.6 |

### 3.4. SEM-EDS

SEM-EDS study on the residue obtained from the experiments with $Na_2CO_3$ at 95 °C, pH 10.6 (Figure 4) and with $NaHCO_3$ at 70 °C, pH 9.0 (Figure 5) was performed, as the residues from these tests contain comparably high amount of $CaF_2$ and lesser amount of $CaCO_3$. Both residues have different ratios of calcite, vaterite and aragonite in the total amount of $CaCO_3$ (Refer Table 2).

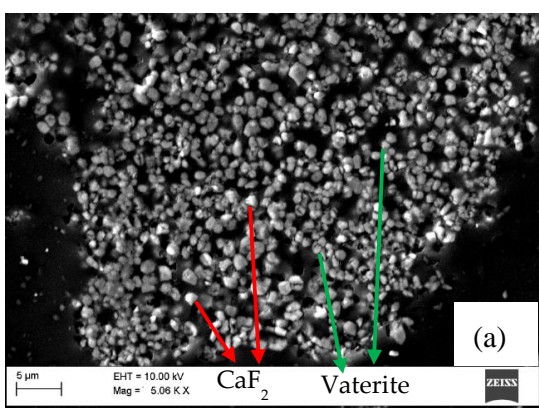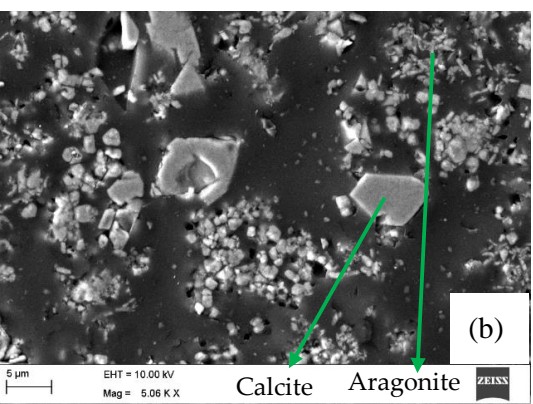

**Figure 4.** SEM image of residue obtained from the experiment with $Na_2CO_3$ at 95 °C and pH 10.6. (**a,b**) show two different regions of the same sample.

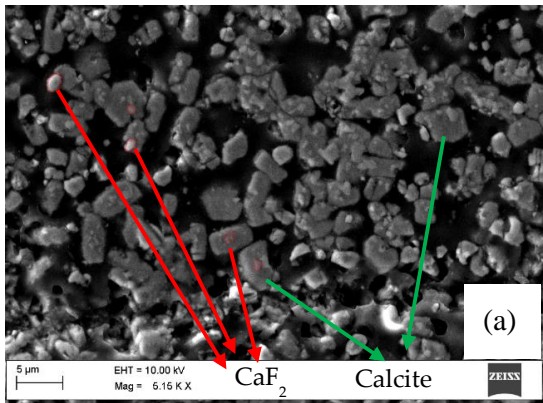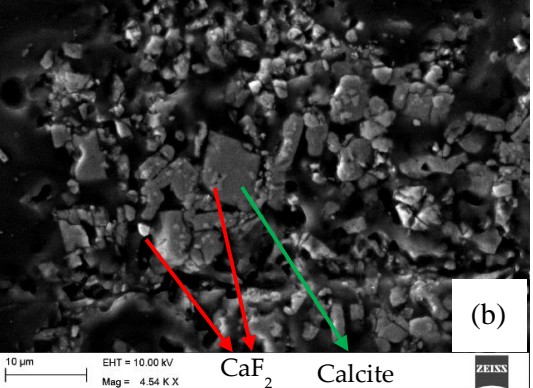

**Figure 5.** SEM image of residue obtained from the experiment with $NaHCO_3$ at 70 °C, pH 9.0. (**a,b**) shows two different regions of the same sample.

In Figure 4a, the residue sample mainly contains ~1 µm particles of $CaF_2$, significantly smaller than 3.37 µm that was the mean diameter of the $CaF_2$ feed. SEM-EDS analyses detected white particles and light gray particles as $CaF_2$, while darker gray particles mainly consist of $CaCO_3$. Spheroidal shaped grains [27] that dominate the residue are vaterite, cf. Figure 4a. Larger particles with sharp edges with a rhombohedral shape are calcite [27], c.f. Figure 4b. Needle-shaped crystals (Figure 4b) were also observed, which correspond to aragonite [27]. As previously mentioned, the dissolution result for this sample shows initially rapid dissolution and, hence, indicates rapid $CaCO_3$ precipitation, until a breakpoint when the fluoride content is unchanged, cf. Figure 1.

Figure 5. shows images of the residue obtained from the experiment with $NaHCO_3$ at 70 °C, pH 9.0. The sample contains slightly larger particles, roughly 2–5 µm and, also, particles with sharp edges and a rhombohedral shape, similar to calcite [27], were quite commonly observed. Large particles of $CaF_2$ were found, indicating unreacted $CaF_2$ due to incomplete dissolution, and some of these seem to be coated with $CaCO_3$, leading to passivation of $CaF_2$ during dissolution reaction. The results of the SEM investigation together with XRD findings indicate that it is not only the temperature and the reagents used but also the polymorphism property exhibited by $CaCO_3$ which has a significant effect on the dissolution behavior of fluoride.

### 3.5. Dissolution Chemistry of CaF₂ at Alkaline pH in Carbonate/Bicarbonate Media

In alkaline condition, $CaF_2$ will be dissolved, as the content of $Ca^{2+}$ in the solution is restricted by precipitation of $CaCO_3$ in the presence of $CO_3^{2-}$ ions [28]. The content of $Na^+$ in the solution remains unchanged during the whole process, as $Na^+$ salts of $F^-$, $CO_3^{2-}/HCO_3^-$ are water soluble. As dissolved $Ca^{2+}$ is removed, the further dissolution of $CaF_2$ is enhanced.

In the aqueous system studied, co-existing equilibriums are:

$$CaF_2 \text{ (s)} \rightleftharpoons Ca^{2+} + 2F^- \tag{3}$$

$$CaCO_3 \text{ (s)} \rightleftharpoons Ca^{2+} + CO_3^{2-} \tag{4}$$

The temperature dependent solubility products for these equilibriums are:

$$K_{sp\text{-}CaF2} = [Ca^{2+}].\ [F^-]^2 \tag{5}$$

$$K_{sp\text{-}CaCO3} = [Ca^{2+}].\ [CO_3^{2-}] \tag{6}$$

When the $Ca^{2+}$ content is lowered below the value corresponding to the solubility product of $CaF_2$, dissolution can continue. Thus, dissolutions of $CaCO_3$ and $CaF_2$ are interdependent, as the equilibria determining their solubility involve concentration of $Ca^{2+}$ in both cases. By keeping the $CO_3^{2-}$ content in the solution high enough, the solubility product for $CaCO_3$ will be exceeded before the one for $CaF_2$ is reached. As $CO_3^2$ ions are consumed, these can be generated again through the equilibrium with present $HCO_3^{2-}$ ions in the solution.

Theoretically, the removal of $Ca^{2+}$ can be enhanced by achieving a higher $CO_3^{2-}$ ion concentration at a higher level of pH; that is, the solubility of $CaCO_3$ [29] decreases with increased temperature as well as pH. Experimentally, it was found that for the dissolution of fluoride, a temperature of 70 °C is preferable at pH 10.6, while 95 °C is preferable at pH 9.0 (Figures 1 and 2). At pH 10.6 the whole carbonate system remains mainly as $CO_3^{2-}$, which facilitates rapid $CaCO_3$ formation. Furthermore, $CaCO_3$ precipitates rapidly at high temperature. The authors Kitano and Okumura showed that a high precipitation rate of $CaCO_3$ favors co-precipitation of $CaF_2$ [19]. At pH 10.6 and 95 °C, it is possible that rapid kinetics due to high temperature, high content of $CO_3^{2-}$ and low solubility of $CaCO_3$ contributes to higher coprecipitation of $CaF_2$ [30]. While at pH 9 the $CO_3^{2-}$ concentration is low and, hence, formation of $CaCO_3$ is also low. So at a higher temperature, less $CaCO_3$ precipitation happens and, hence, less coprecipitation of $CaF_2$ is found. During precipitation of $CaCO_3$ encapsulation of $CaF_2$

particles is also possible, which leads to passivation of the reaction, which was observed in SEM studies. In such conditions, total fluoride dissolution will be lower.

According to Norio Wada et al., in systems of soluble calcium nitrate-sodium carbonate $(Ca(NO_3)_2\text{-}Na_2CO_3)$ and soluble calcium chloride-sodium carbonate $(CaCl_2\text{-}Na_2CO_3)$, aragonite precipitation is favored at high temperature and in the presence of divalent cations [31] such as $Zn^{2+}$, $Fe^{2+}$, $Mg^{2+}$. Hence, during leaching of WO, DLWO and other zinc-containing dust material with $Na_2CO_3/NaHCO_3$ at higher temperatures, aragonite formation and co-precipitation of $CaF_2$ may occur. Complex interplay of both thermodynamic and kinetic factors affects $CaCO_3/CaF_2$ equilibrium; to elucidate, it is influenced by factors such as temperature, retention time, pH, reagents used, stirring conditions, etc. This leads to polymorphism of $CaCO_3$ [16,17,26], which influences dissolution behavior of $CaF_2$ by either passivating it or co-precipitating it.

## 4. Conclusions

- This experimental and thermodynamic dissolution study on $CaF_2$ showed that the dissolution chemistry is similar irrespective of the use of 150 mM of either $Na_2CO_3$ or $Na_2HCO_3$ as reagent. The following conclusions are drawn from this work:

  o Highest dissolution of CaF2 at pH 9 is achieved at 95 °C.
  o Highest dissolution of CaF2 at pH 10.6 is achieved at 70 °C.
  o An initially strong positive effect on the dissolution of CaF2 is observed at a higher temperature.
  o According to the experimental results, the complete dissolution of CaF2 was not achieved in any of the dissolution conditions studied.
  o According to the thermodynamic calculations, complete dissolution of CaF2 is possible in all the dissolution conditions studied.

- XRD-Rietveld refinement and SEM-EDS investigations showed that a mixture of polymorphs of $CaCO_3$, namely calcite, vaterite and aragonite, are formed in different ratios for different reaction conditions.

  o The type of polymorphs formed has an impact on the fluoride dissolution.
  o Higher temperature and higher alkaline pH cause rapid $CaCO_3$ precipitation, forming aragonite and higher coprecipitation of $CaF_2$.

- In dissolution of $CaF_2$ to $Ca^{2+}$ and $F^-$, control of aqueous $Ca^{2+}$ concentration by $CO_3^{2-}$ concentration plays an important role. Consumption of $Ca^{2+}$ by $CaCO_3$ precipitation is affected by multiple factors including temperature and pH, which that influence the $CO_3^{2-}$ concentration and the type of $CaCO_3$ polymorph (calcite, vaterite and aragonite) formed.

**Author Contributions:** Conceptualization, C.S., L.S.Ö., S.S.; methodology, S.S. and L.S.Ö.; formal analysis, S.S.; investigation, S.S.; data curation, S.S.; writing—original draft preparation, S.S.; writing—review and editing, S.S., C.S. and L.S.Ö.; supervision, C.S., L.S.Ö. and F.E.; project administration, C.S., L.S.Ö.; funding acquisition, C.S. All authors have read and agreed to the published version of the manuscript.

**Funding:** This research was funded by Boliden Commercial AB through Bolidenpaketet.

**Acknowledgments:** The authors are thankful to Mehdi Parian for the kind help in recording XRD and SEM images. Fruitful discussions with Boliden are gratefully acknowledged.

**Conflicts of Interest:** The authors declare no conflict of interest.

# Appendix A

*Appendix A.1. pH*

The literature [3,4] suggests having pH at 9. The self-generated pH of the slurry of water, Waelz oxide (WO) and 16 g/L $Na_2CO_3$ was ~9.0 and not ~11. Since zinc oxide in WO will get converted to amphoteric $Zn(OH)_2$ in aqueous solution, acting as buffering reagent, converting $Na_2CO_3$ added to $NaHCO_3$ as below reaction:

$$H_2ZnO_2 + Na_2CO_3 \rightleftharpoons NaHZnO_2 + NaHCO_3 \tag{A1}$$

$$NaHZnO_2 + Na_2CO_3 \rightleftharpoons Na_2ZnO_2 + NaHCO_3 \tag{A2}$$

The solubility product of $Zn(OH)_2$ is $3 \times 10^{-16}$ Hence the pH of saturated $Zn(OH)_2$ solution can be calculated as:

In aqueous solution, $Zn(OH)_2$ dissociates as:

$$Zn(OH)_2(s) \rightarrow Zn^{2+}(aq) + 2OH^-(aq) \tag{A3}$$

Hence,

$$K_{sp} = [Zn^{2+}][OH^-]^2 \tag{A4}$$

Suppose the solubility of $Zn(OH)_2$ is $z$, and it completely dissociates in aqueous solution. Then at the equilibrium, concentration of $Zn^{2+}$ will be $z$ while that for $OH^-$ will be $2z$. By applying these values into the $K_{sp}$ equation,

$$K_{sp} = [Zn^{2+}][OH^-]^2 = z.(2z)^2 = 4z^3 = 3.0 \times 10^{-16} \tag{A5}$$

$$z^3 = 7.5 \times 10^{-17} \tag{A6}$$

$$z = 4.2 \times 10^{-6} \tag{A7}$$

$$[OH^-] = 2z = 8.4 \times 10^{-6} \tag{A8}$$

$$pOH = -\log[OH^-] \approx 5 \tag{A9}$$

Therefore, pH = 14 − pOH = 9 is obtained when 16 g/L $Na_2CO_3$ is used for washing of WO.

*Appendix A.2. Thermodynamic Calculations*

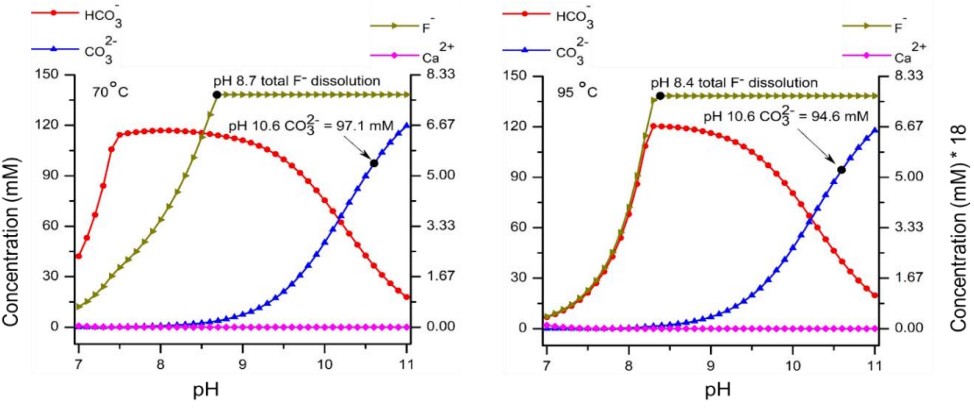

**Figure A1.** Thermodynamic calculation with 150mM $Na_2CO_3$ and 300 mg/L $CaF_2$ from pH range 7–11 and temperature (70 and 95 °C).

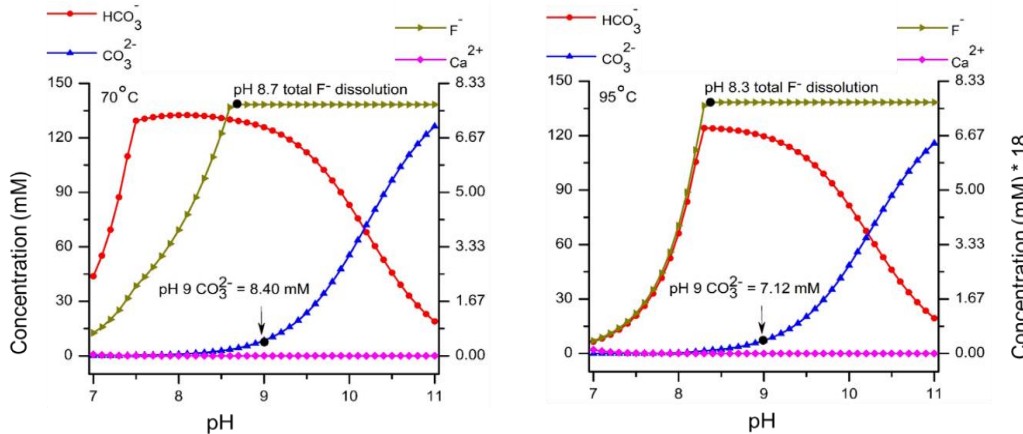

**Figure A2.** Thermodynamic calculation with 150 mM NaHCO$_3$ and 300 mg/L CaF$_2$ from pH range 7–11 and temperature (70 and 95 °C).

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
