# Peer review of "Experimental Study on the Dissolution Behavior of Calcium Fluoride"

_metals, doi:10.3390/met10080988_

Round 1
Reviewer 1 Report
Dear Sir,
The manuscript is presenting some insights in possible pathways of eliminating CaF2 through dissolution in aqueous Na2CO3 or NaHCO3 as means of purifying Waelz oxide before further refining. The manuscript is well written and the experimental part is well conducted and explained. However, the authors failed to mention previous researches in the same area and compare their own results with these previous studies. For example, solubility experiments of CaF2 in Na2CO3 and NaHCO3 solutions were already carried out by Rao et al. as early as 1993 (Environmental Geology, 1993, 21, 84-89). Interference between CaF2 and bicarbonate were studied because of their possible encounter in oral hygiene products (Peres et al., Effect of bicarbonate on fluoride reactivity with enamel, Rev. odonto ciênc. 2009, 24(1), 6-9; Sun and Chow, Preparation and Properties of Nano-sized Calcium Fluoride for Dental Applications, Dent Mater. 2008; 24(1): 111–116). Solubility of CaF2 in water or saline solutions (including seawater) was also investigated (Garand and Mucci, The solubility of fluorite as a function of ionic strength and solution composition at 25C and 1 atm total pressure, Marine Chemistry 91 (2004) 27– 35; Tropper and Manning, The solubility of fluorite in H2O and H2O–NaCl at high pressure and temperature, Chemical Geology 242, 2007, 299–306).
If the studied process is destined to the removal of fluorite from Waelz oxide, how would the latter behave when submitted to the same solubilization conditions? The answer is given in the authors’ own previous paper (Sar et al., Metals, 2019,9, 361; doi:10.3390/met9030361 – ref. 7): “The dust known as Waelz oxide (WO) contained up to 60% zinc and was further purified for dehalogenation using a double-leaching process with sodium carbonate (Na2CO3) [17,18]. The double-leached WO (DLWO) contained61–66% zinc, 7-9% lead, 3.5-5.5% iron and the content of the halogens was reduced to <0.1% for Cl and <0.15% for F in the leaching process[2].”
I assume that since their previous study, the authors aimed to a more advanced removal (up to complete removal) of CaF2 from WO, hence the present study.
Maybe that briefly remembering these previous studies, along with a short discussion on WO’s behavior in similar conditions, would give a new layer to the present manuscript?
Minor comments:
- Please use throughout the manuscript “L” as international symbol for “liter”
- Please revise the references list for uniformity
Otherwise, the study is well-conducted and the information presented therein is worthwhile for publication. Therefore, I suggest acceptance after minor corrections.
Reviewer 2 Report
The current investigation concerns the dissolution behavior of calcium fluoride and also some aspects of precipitation of the produced products in the solution. The research approach and the interpretation of the results are in general unambiguous and the analysis of the results has been carried out thermodynamic principles utilizing modern pieces of equipment. This reviewer recommends that the paper be published with minor revision.
Appropriate paragraphing is needed in the Abstract.
Line 234 – “Thermodynamically stable crystals are formed when crystallization process is kinetically controlled.” This statement is unclear.
Some explanation should be given the choice of temperatures of 70 and 95 instead of other temperatures. Similarly, the significance of the choice of the pH of 9 and 10.6 used in the study should also be explained. For example, the study can be conducted over a range of temperatures and pH.
Eqs. 3 and 4 do not represent the complete equilibrium situations in the aqueous solution. For example, the presence and the potential effect of HCO3- and HF on the overall equilibrium should be discussed.
The study has been carried out in terms of thermodynamic consideration on dissolution and precipitation but very little kinetic behaviors are dealt with throughout the study. However, the authors have made arguments in terms of thermodynamics as well as kinetics.
